# Long-Term Calculation of Predicted Environmental Concentrations to Assess the Risk of Anticancer Drugs in Environmental Waters

**DOI:** 10.3390/molecules27103203

**Published:** 2022-05-17

**Authors:** Pol Dominguez-García, Marta Gibert, Sílvia Lacorte, Cristian Gómez-Canela

**Affiliations:** 1Department of Analytical and Applied Chemistry, School of Engineering, Institut Químic de Sarrià-Universitat Ramon Llull, Via Agusta 390, 08017 Barcelona, Spain; poldominguezg@iqs.url.edu (P.D.-G.); martagiberta@iqs-blanquerna.url.edu (M.G.); 2Institute for Environmental Assessment and Water Research (IDAEA-CSIC), Jordi Girona 18, 08034 Barcelona, Spain; slbqam@cid.csic.es

**Keywords:** anticancer drugs, PECs, wastewater, river water, fate

## Abstract

This study reports the consumption data for 132 anticancer drugs in Catalonia (NE Spain) during the period of 2013–2017 and calculates the predicted environmental concentrations (PECs) in wastewater effluents and rivers. This long-term analysis can determine the evolution of drugs present in the environment according to prescriptions and serve as an adequate tool to determine their presence and impact. Data showed that out of 132 compounds prescribed, 77 reached wastewater effluents, which accounted for the most consumed, those excreted in the highest doses, and the least biodegradable. Once diluted in receiving river waters, only mycophenolic acid and hydroxycarbamide had PEC values higher than 10 ng L^−1^, which is the value set by the European Medicines Agency (EMA) to carry out further risk assessment. It was also observed that compounds present in river water are those that can pose a high risk, given their persistence and capability to bioaccumulate. Therefore, this study shows that the estimation of PEC, together with physico-chemical properties of detected compounds, is a useful tool to determine the long-term presence and fate of this new class of emerging contaminants.

## 1. Introduction

Anticancer drugs are a family of pharmaceuticals classified as antineoplastic and immunomodulating agents. These drugs are consumed worldwide on a regular basis with prescription levels varying according to the country. Although there are still difficulties in obtaining country-based consumption values, some studies have published the annual consumptions. Besse et al. reported data from the French Health Products Safety Agency for cytotoxic and cytostatic drugs and show that the national consumption amount in 2008 was 17.5 t and included 88 drugs [1]. Consumption amounts in Germany for antineoplastic drugs were 20.7 t and included 102 drugs, according to data from 2012 [2]. Consumption amounts in north-west England were 0.6 t, consisting of 46 drugs [3]; in Portugal 5.8 t of 171 drugs, in 2015 [4]; and in Spain 25 t of 78 drugs, in 2015 [5]. In all these studies, the consumption per capita varied according to compounds, but generally ranged between 1–200 µg/capita/day. New anticancer drugs are annually placed in the market to combat cancer, one of the main causes of mortality worldwide [6]. As other classes of pharmaceuticals, after being administered, anticancer drugs enter the sewage system and end up in river water as a result of high excretion rates, poor degradability in wastewater treatment plants (WWTPs), and high environmental persistence of some of them [3]. The result is that residues are systematically detected in river waters worldwide [7], with areas with water scarcity showing higher levels due to the poor dilution capacity of receiving waters [8]. These compounds are especially relevant from a toxicological point of view because of their mode of action. Being bioactive substances, their presence in environmental waters can affect aquatic organisms at different organizational levels [9,10] and can have notorious effects on the long term [11,12]. There are two methods of studying the presence and fate of anticancer drugs in surface waters: monitoring [13,14,15,16,17] and prediction of the levels in different water bodies [18,19]. Monitoring is probably the most accurate procedure to determine the concentrations of these compounds in water, but faces several limitations because: (i) anticancer drugs are not regulated and thus there is no need to determine their levels in surface waters; (ii) methods can only cope with a few compounds (e.g., 10–30) out of those administered, basically due to analytical limitations; and (iii) available monitoring data and time trends is scarce, mainly because of the cost of monitoring. Therefore, it is very difficult to determine the occurrence of anticancer drugs in water with regard to the main risk compounds, and their evolution over time cannot be assessed with accuracy. In the last few years, estimations of the predicted environmental concentrations (PECs) have gained importance because of the ability to (i) semiquantitatively determine the concentration of a large number of anticancer drugs in a water body, based on consumption data and (ii) the possibility of evaluating their presence in different areas and determining the time trends. PECs are useful tools to prioritize compounds with higher chances of being present in waters, and, thus, to further implement monitoring programs to evaluate their presence and impact [20].

In a previous study, authors evaluated the consumption and PEC values of a set of 132 anticancer drugs prescribed in Catalonia during the period of 2010–2012, which permitted the prioritization of the main compounds in water [19]. Upon monitoring, mycophenolic acid was detected for the first time in river water [21], showing the importance of PEC for identifying contaminants of emerging concern. Thus, the aim of this study is to demonstrate the usefulness of PEC calculations for the long-term estimation of 132 anticancer drugs in wastewater and rivers over the period 2010–2017 and in prioritizing the most prevalent, which should be considered in monitoring studies. For compounds with the highest PEC values, a risk assessment study was conducted in order to predict their possible effects in the aquatic environment.

## 2. Results and Discussion

### 2.1. Consumption of Cytostatic Drugs in Hospitals and Pharmacies from Catalonia

According to the data provided by CatSalut, the total consumption of anticancer drugs ranged from 4.9 t to 5.8 t during the period 2013–2017. Figure 1 displays the consumptions of the ATC groups related to cytostatic drugs administered in pharmacies and hospitals from Catalonia during the period of 2013–2017. The ATC groups with higher levels of consumption were L04 Immunosuppressants, defined as agents that completely or partly suppress one or more factors in the immune system and L01 Antineoplastic agents, which interfere with cancer cell’s ability to grow and spread in a variety of ways. L04 levels increased from 2.57 t year^−1^ (in 2013) to 3.10 t year^−1^ (in 2017), see Figure 1. The same trend was reported by L01 drugs; their levels increasing from 1.64 t year^−1^ (in 2013) to 1.87 t year^−1^ (in 2017). The next most consumed groups were L02 (endocrine therapy) and H02 (Corticosteroids for systemic use), which include prednisone, used in the treatment for a variety of cancers, such as leukemia, lymphoma, and multiple myeloma, to treat the nausea and vomiting associated with some chemotherapy drugs and to stimulate appetite in cancer patients with severe appetite problems. The levels of both groups were constant over the years studied, being 0.42 t per year^−1^ (in 2013) and 0.43 t per year^−1^ (in 2017) for L02 and 0.24 t per year^−1^ (in 2013) to 0.35 t per year^−1^ (in 2017) for H02 (Figure 1). Appendix A shows the consumptions (in t) of cytostatics in pharmacies and hospitals, as classified by ATC codes.

Table 1 summarizes the 15 most consumed cytostatic drugs in the 2010-2017 period considering the previously published data on cytostatic drugs consumption between 2010 and 2012 [19]. Data show that mycophenolic acid, an immunosuppressant medication used to prevent rejection following organ transplantation and to treat autoimmune conditions such as Crohn’s disease and lupus, was the most consumed cytostatic drug throughout the period studied. Moreover, its consumption (in µg inhab^−1^ day^−1^) increased considerably between the years analyzed, with 5093 g day^−1^ in 2010 and 7331 µg inhab^−1^ day^−1^ in 2017. The next most consumed cytostatic drug was hydroxycarbamide, also known as hydroxyurea, which is used in sickle-cell disease, chronic myelogenous leukemia, cervical cancer, and essential thrombocythemia. The consumption of hydroxycarbamide, a medication used in sickle-cell disease, chronic myelogenous leukemia, cervical cancer, and essential thrombocythemia also increased throughout the period studied, from 1598 g day^−1^ (in 2010) to 2217 g day^−1^ (in 2017). Conversely, the consumption of capecitabine, which is a pharmaceutical used for colon cancer treatment, has decreased throughout the last few years. Thus, for this drug, the maximum levels consumed were in 2011, at 2273 g day^−1^ [19], decreasingly up to 1607 g day^−1^ in 2014, and then its level started to increase again up to 1899 g day^−1^ in 2017 (Appendix A). The next most consumed cytostatic compound was prednisone, a pharmaceutical used for arthritis treatment and breathing problems, with an increase in its consumption to around 355 g day^−1^ from 2010 to 2017. In 2010, the consumption was 608 g day^−1^ and increased slightly until 2017 with a consumption rate of 963 g day^−1^. Like prednisone, azathioprine is used for the prevention of organ rejection and also had an increase in its consumption to levels of 815 g day^−1^ in 2017. However, the opposite happened with megestrol, a pharmaceutical used to treat loss of appetite; a consumption rate of 619 g day^−1^ was appraised in 2010 but it had decreased remarkably by 2015, with the lowest consumption rate of 391 g day^−1^. However, consumption increased again to 447 g day^−1^ in 2017 (Appendix A). The drugs imatinib, used for leukemia treatment, and abiraterone, used for prostate treatment, increased slightly, and ciclosporin, which is used for certain types of skin cancer, maintained its levels throughout the years studied. 

Bicalutamide, which is used to treat prostate cancer, had a notable decrease in its level of consumption. Levels remained stable in 2010 and 2011, with consumption rates of 303 and 302 g day^−1^ [19], respectively, but from 2012 to 2017 decreased considerably, with a consumption rate of 166 g day^−1^ in 2017. Tamoxifen and gemcitabine, used for breast cancer treatment, maintained their consumption patterns throughout the years studied (Appendix A). On the other hand, nilotinib increased its consumption from 42 g day^−1^ in 2010 [19] to 93 g day^−1^ in 2017 (see Table 1). Finally, flutamide is a special case, due to its higher consumption in 2010 (259 g day^−1^) [19] and its poor consumption in 2017 (47 g day^−1^). The consumption of pharmaceuticals by the global population varies between countries and its levels are expected to grow as the populations age; levels also depend upon polymedication. Furthermore, new advances in cancer therapeutics have been proven in the last few years, and new medicines for cancer treatment have been recently approved. In the period of 2013 to 2018, 63 new active substance were approved for cancer treatment, according to the Institute for Human Data Science (IQVIA) [22]. This is in accordance with the generally increasing consumption of most of the cytostatic drugs found in Catalonia.

### 2.2. PECs in Wastewater Effluents

Predicted concentrations in wastewater effluents (PEC_wwtp_), which represent the amount expected in WWTP effluents, were calculated for all cytostatic drugs consumed in Catalonia during the period of 2013–2017. One relevant parameter in the PEC calculation is the fraction excreted and the fraction of drugs eliminated in WWTP. For the former, excretion depends on many factors, such as age, health, and the condition of the patient and several other values can be considered. This can produce an uncertainty in the calculation. In this study, the excretion data used correspond to published data or data estimated with EPIsuite^TM^ 4.11 software (“EPI SuiteTM-Estimation Program Interface”, U.S [23]. Environmental Protection Agency, Syracuse, NY 13212), and it is our consideration that drugs will never be excreted 100% (which would be the worst-case scenario) as drugs perform necessary activity in the body and at least 50% are excreted in a few hours [24]. Appendix A displays the levels of excretion and the percentage of removal for each cytostatic drug. Regarding the F_wwtp_, the situation can vary in different countries and thus it is useful to have information on the wastewater management in the study area. In Spain, more than 90% of the urban wastewater is treated, with 37% receiving secondary treatment, and 51% given additional tertiary treatment [25] which represents the maximum level of elimination of organic matter and contaminants. For this reason, F_wwtp_ has been chosen, considering that in the secondary treatment, drugs are biodegraded. Appendix A displays, for each year, the relation between the consumption levels and the PEC levels of 132 cytostatic drugs from Catalonia, according to CatSalut data in 2013, 2014, 2015, 2016, and 2017. Table 2 reports the cytostatic drugs with highest levels of PEC_wwtp_. Regarding PECs in wastewater effluents, cytostatic drugs with higher levels were mycophenolic acid, hydroxycarbamide, capecitabine, bicalutamide, imatinib, prednisone, and leflunomide. Mycophenolic acid was the cytostatic drug with highest values of PEC_wwtp_ with levels between 2224 ng L^−1^ (in 2013) and 2750 ng L^−1^ (in 2017). In a previous study on the estimation of PECs in Catalonia during 2010–2012, authors reported a mean value of 2008 ng L^−1^ [19]. Following this, hydroxycarbamide had a PEC_wwtp_ between 892 ng L^−1^ (in 2013) and 1098 ng L^−1^ (in 2017), see Table 2. Franquet-Griell et al. reported a mean PEC_wwtp_ of 832 ng L^−1^ [19], being a more similar value than the values reported in the present study. These values are also in accordance with those reported in France in 2012, with values of PEC_wwtp_ of 781 ng L^−1^ [1]. 

Capecitabine and bicalutamide had similar values of PEC_wwtp_, with levels between 165 ng L^−1^ (in 2013) and 179 ng L^−1^ (in 2017) and between 132 ng L^−1^ (in 2013) and 89.6 ng L^−1^ (in 2017), respectively. In comparison with the previous study of the estimated concentration of cytostatic drugs in Catalonia, values of PECs for capecitabine and bicalutamide have decreased throughout the last few years, consisting of a maximum value of PEC_wwtp_ in the period 2010–2012 with mean values of 201 and 156 ng L^−1^, respectively [19]. In 2013, Johnson et al. studied the PECs of the cytostatic drugs cyclophosphamide, carboplatin, 5-fluorouracil, and capecitabine throughout the sewage effluents and surface waters of Europe. Authors reported the highest PECs of capecitabine in wastewater effluents in the Czech Republic, Denmark, and the Netherlands, with values of 87 ng L^−1^, 48 ng L^−1^, and 46 ng L^−1^, respectively [26], lower to our results. 

According to its constant consumption rates throughout the years studied, imatinib registered PEC_wwtp_ values between 65.1 ng L^−1^ (in 2013) and 70.7 ng L^−1^ (in 2017). Following, prednisone and leflunomide had PECs between 43.2 ng L^−1^ (in 2013) and 63.1 ng L^−1^ (in 2017) and between 42.2 ng L^−1^ (in 2013) and 51.2 ng L^−1^ (in 2017); see Table 2. Finally, the rest of the pharmaceuticals with PECs values in differing orders of magnitude were between 10.4 ng L^−1^ (in 2013) and 19.3 ng L^−1^ (in 2017).

As can be seen, from 2013 to 2017, most PEC values had a tendency to increase. This is in accordance with the increasing consumption of cytostatic compounds in the last few years [22].

### 2.3. PECs in Rivers

PEC_river_ (in ng L^−1^) was calculated by considering the dilution factor of WWTP to surface water and represents the concentration expected in rivers. EMA proposed the calculation of PECs and suggested evaluating their presence and effects when PEC values in surface water are equal or above 10 ng L^−1^ [27]. Table 2 displays the PEC_river_ of the prioritized cytostatic drugs. According to EMA guidelines, two cytostatic drugs (mycophenolic acid and hydroxycarbamide) had PEC levels higher than 10 ng L^−1^; the threshold value for the environmental risk assessment of pharmaceuticals [27]. Mycophenolic acid was estimated in the range of 86 ng L^−1^ (2013) to 106 ng L^−1^ (2017); and hydroxycarbamide in the range of 34 ng L^−1^ (2013) to 42 ng L^−1^ (2017). In comparison with the levels reported in a previously published study [19], the PEC_river_ value of mycophenolic acid has increased. The mean value of mycophenolic acid from 2010 to 2012 was 77 ng L^−1^ which is considerably lower than the results obtained in 2017. Additionally, the PEC_river_ of hydroxycarbamide has increased from the mean value of 32 ng L^−1^ in the previous study from the 2010–2012 period, to more than double in 2017. These results clearly show a tendency towards levels increasing and emphasize the potential environmental impacts in the upcoming years.

Regarding the other cytostatic drugs, bicalutamide, capecitabine, imatinib, prednisone, leflunomide, and pazopanib, had average values of 6.4, 4.2, 2.6, 2.0, 1.8, and 1.1 ng L^−1^. Comparing mean values from a previous study [19], there are some relevant PEC river increases, such as pazopanib, which had a mean value of 0.08 ng L^−1^ in 2010–2012 and 1.1 ng L^−1^ in the 2013–2017 period. This indicates the possible future concern if any of them reach the concentration of 10 ng L^−1^ proposed by EMA. On the other hand, there are also decreases of PEC mean values. Capecitabine had a mean value of 7.8 in 2010–2012 and a mean value of 4.2 in the 2013–2017 period, and so far pose no environmental threat. Finally, the rest of the 132 cytostatic compounds had PEC values lower than 1 ng L^−1^.

### 2.4. PECs vs. MECs

There are several criteria for PECs vs. MECs estimation. Coetsier et al. proposed a method to clarify whether PECs tend to be underestimated or overestimated regarding MECs [28]. Values between 0.2 < PEC/MEC < 1 should indicate acceptable results lightly underestimated, 1 < PEC/MEC < 4 values indicate acceptable results lightly overestimated, and, finally, 4 < PEC/MEC < 8 values indicate sever overestimation.

PEC/MEC estimations were calculated from the highest PEC values of all cytostatic compounds using MEC values from the bibliography. Table 3 shows the PEC/MEC values obtained from the most detected cytostatic drugs. Ratios from mycophenolic acid, which had the highest PEC value, had a PEC/MEC value of 0.2 in river water, calculated from the MEC value that was reported in a previously published paper about cytostatic drugs in the Besòs River [16]. That value indicates an underestimation following the PEC/MEC ratios proposed by Coestier et al. Capecitabine had PEC/MEC ratios of 6.2 in wastewater influent, 21.6 and <11.1 in wastewater effluent, and gave overestimated results from MEC values reported in Negreira et al., who were studying the presence of drugs and metabolites in municipal and hospital wastewaters in 2013 and 2014 [29,30]. Imatinib had a PEC/MEC ratio of < 0.4 in wastewater influent and < 0.4 in river water and gave underestimated results in surface water from Catalonia (Table 3) [29]. Finally, prednisone had PEC/MEC ratios of < 4.4 in surface water and gave a slightly overestimated PEC/MEC ratio, calculated from MEC values reported in a previously published paper on the occurrence of cytostatic compounds [13]. 

Thus, the PEC/MEC ratio is a primary tool to predict cytostatic compounds in wastewaters (influents and effluents) and river water that serve to prioritize pharmaceutical compounds that should be monitored in the aquatic environment.

### 2.5. Risk Assessment

Risk assessment was performed to determine the impact of cytostatic drugs on aquatic resources. PEC values from 2013 to 2017 were used and EC_50_ and LC_50_ values were obtained from the bibliography. Experimental bibliographical data regarding EC_50_ and LC_50_ should be used carefully as there are multiple variations of values between different studies. Only compounds with PEC values higher than 0.1 ng L^−1^ in river waters were considered when calculating RQ. Although EMA indicated a limit of 10 ng L^−1^ as an environmental threat, values of 0.1 ng L^−1^ were considered, as cytostatic drugs can have higher chronic effects than expected (see Table 4) and can become an environmental risk in following years. Table 4 shows the values of the RQ calculated from toxicological bibliographical data. All compounds had a low risk (< 1), with bicalutamide had RQ value of 0.004, calculated using the *D. magna* 24 h toxicity test (1 mg mL^−1^), followed by mycophenolic acid with a RQ value of 0.001, also calculated using the *D. magna* 24 h toxicity test (> 100 mg mL^−1^), paclitaxel, with a value of 0.0008, calculated using the *D. magna* 24 h toxicity test (0.74 mg mL^−1^); hydroxycarbamide with a value of 0.0004, also calculated using the *D. magna* 24 h toxicity test (> 100 mg mL^−1^); cyproterone and imatinib, both with a value of 0.0002, calculated using the *D. magna* 48 h toxicity test (2.4 and 2.58 mg mL^−1^, respectively); and megestrol, with a value of 0.0001, calculated using the *D. magna* 48 h toxicity test (5 mg mL^−1^). At this point, the rest of RQ values were lower than 0.0001 and differed considerably from the value of 1, showing no risk for the environment based on the maximum probable risk for ecological effects on contaminated water [31]. 

Comparing these results with the previous study in 2015 [19], the RQ for mycophenolic acid increased from 0.0008 (in the 2010–2012 period) to 0.001 (in the 2013–2017 period). Other cytostatic drugs with high mean PECs values, such as hydroxycarbamide (38 ng L^−1^) and capecitabine (6.4 ng L^−1^), had very low RQ values because their EC_50_ concentrations were very high and thus produce a low RQ value.

## 3. Materials and Methods

### 3.1. Consumption Data

According to the WHO Collaborating Center for Drug Statistics Methodology, cytostatic drugs are classified using the anatomic therapeutic chemical (ATC) code, which considers the anatomical and pharmacological properties and allows further subclassifications according to therapeutic uses and chemical structure [42]. There are 378 approved drugs for that purpose, which include antineoplastic agents (L01, 252 drugs), endocrine therapy (L02, 13 drugs), immunostimulants (L03, 48 drugs), immunosuppressants (L04, 71 drugs), sex hormones (G03, 142 drugs), and corticosteroids (H02, 23 drugs); 171 drugs are in multiple categories) The ability to obtain consumption data at a country level or more regionally is key for estimating the PECs. In the present study, consumption data for cytostatic drugs were provided by the Catalan Health Service (CatSalut), requested through their ATC code. All antineoplastic pharmaceuticals from the period of 2013 to 2017 from hospitals and pharmacies consisted in total of 132 cytostatic drugs, which are the prescribed drugs in the period studied. Hospital data were given as pill numbers or active molecule formulation (activities). From the concentration of the activity of each compound, annual consumption was calculated in g year^−1^. 

### 3.2. Calculation of PECs

PECs were calculated for each drug according to Equations (1) and (2). Equation (1) displays the calculation of the PECs in the wastewater treatment plant effluents (WWTP_eff_), and Equation (2) shows the calculation of PECs in rivers:(1)$$\mathrm{PEC}\;\mathrm{WWTP}\;\mathrm{eff}=\frac{\mathrm{consumption}\;\times \;\mathrm{Fexcreta}\;\times \left(1-\mathrm{Fwwtp}\right)}{W\;\times \;\mathrm{inhabitants}\;}\;\times \;10^{9}$$
where consumption (g day^−1^) is the amount of each cytostatic drug administrated in Catalan hospitals and pharmacies. These values include all the pharmaceutical forms administered for each drug. F_excreta_ is the excreted fraction of the unchanged drug, considering both urine and feces. For those compounds whose values could not be found, a default value of 0.5 was applied. F_wwtp_ refers to the removal fraction in WWTP. When different data were obtained in the bibliography, the lowest value was the one used, and for the ones that the value could not be found, the worst-case scenario was contemplated, so the used value for this parameter was 0. Water consumption (W, L inhabitant^−1^ day^−1^) is the water consumption per inhabitant per day (about 130.9 L inhab^−1^ d^−1^ in Catalonia); and finally, the inhabitants are the population in Catalonia, which, using data from 2012, consists of 7.570.908 inhabitants.
(2)$$\mathrm{PEC}\;\mathrm{river}=\frac{\mathrm{consumption}\;\times \;\mathrm{Fexcreta}\;\times \left(1-\mathrm{Fwwtp}\right)}{W\;\times \;\mathrm{inhabitants}\;\times \;\mathrm{DF}}\times 10^{9}\;$$

Regarding Equation (2), the dilution factor (DF) should be considered. The DF is the most critical parameter in the estimation of PECs in rivers, referring to the dilution of WWTP effluents to surface waters. Changes in this value can vary the results more than 100-fold. The European Medicines Agency (EMA) proposes a DF of 10 [27]. However, this value can vary by orders of magnitude when considering the different riverine regimes around the world. This can be overcome by using the DF values obtained by Keller and colleagues in 2014, who calculate the median dilution factor for each country, based on a model that divides the terrestrial surface into fractions of 0.5° × 0.5° (equivalent to 55 × 55 km at the Equator) and take into consideration the river flows, the water consumption and the population for each specific country [43]. The results obtained vary between 0.0050 in Qatar and 94,463 in Suriname. Spain received a value of 25.92, greater than the default value of the EMA, e and this value was used to calculate the PECriver in Spain [19].

### 3.3. Estimation of Risk Quotient

The risk quotient (RQ) expresses the risk posed by a chemical to the environment or organisms [19]. It is calculated using the result of the predicted effect concentration (PECs) and the predicted no effect concentration (PNECs). RQ was calculated according to Equation (3).
(3)$$\mathrm{RQ}=\frac{\mathrm{PEC}}{\mathrm{PNEC}}=\frac{\mathrm{PEC}}{{\mathrm{EC}}_{50/f}}$$
where PNEC can be estimated as the toxicological relevant concentration (EC_50_ or LC_50_) and a security factor (*f* = 1000) is used for the compensation of the scarce chronic toxicity, as PNEC values refer to acute toxicity of the organisms [44]. PNEC values were extracted from the bibliography from two different aquatic organisms (Daphnids and fish).

Results were interpreted following the maximum probable risk for ecological effects from contaminated water [31], where RQ < 1 indicates no significant risk, values between 1 ≤ RQ < 10 indicate a small potential for adverse effects, values between 10 ≤ RQ < 100 indicate the potential for adverse effects and, finally, RQ ≥ 100 indicates the potential for adverse effects.

## 4. Conclusions

Consumption data for all cytostatic compounds were studied in Catalonia from 2010 to 2017. Data showed that the L04 group (immunosuppressants) were the most consumed cytostatic drugs in the period studied, and in 2017 the highest consumption values were observed (3.1 t). PEC values were estimated in WWTPs and river waters. The 15 cytostatic drugs with the highest PECs are in the range between 10 to 2500 ng L^−1^ in WWTPs and between 0.4 to 100 ng L^−1^ in river water. The PEC values in WWTPs were estimated by considering the excreted fraction of each different compound and their removals in the treatment plants. In river water, only 2 cytostatic drugs out of 132 (mycophenolic acid and hydroxycarbamide) had PECs over 10 ng L^−1^, which is the EMA threshold for considering potential environmental risk. The PEC/MEC ratio was studied for cytostatic compounds with high PEC values. This ratio calculation enabled the assessment of whether the prediction was overestimated or underestimated and provided a chance of validating the PEC calculation. Risk quotient was calculated from PEC values higher than 0.1 ng L^−1^. Data showed no environmental risk in any of the cytostatic compounds, as all values were lower than 1, even though PEC values increased from 2013 to 2017, indicating that they may become a threat in the future. Furthermore, ecotoxicological data for some cytostatic compounds was scarce. 

## Figures and Tables

**Figure 1 molecules-27-03203-f001:**
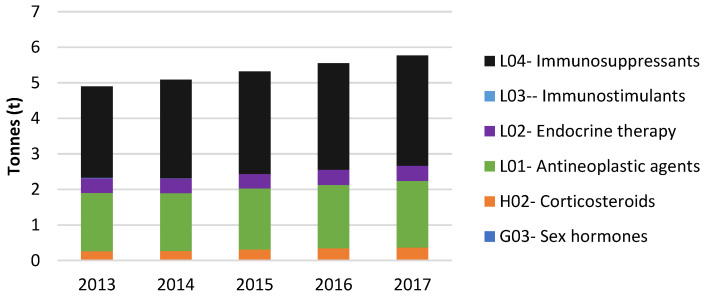
Annual consumption (in t) of cytostatic drugs in pharmacies and hospitals classified by the ATC code.

**Table 1 molecules-27-03203-t001:** Consumption (in g day^−1^) of the 15 most consumed cytostatic drugs in the period of 2013–2017. * Data of previous published paper from Franquet-Griell et al, 2015. [19].

				Consumption (g day^−1^)				
Cytostatic Drug	2010 *	2011 *	2012 *	2013	2014	2015	2016	2017
Mycophenolic acid	5093	5408	5554	5928	6456	6788	7036	7331
Hydroxycarbamide	1598	1742	1698	1801	1906	2007	2123	2217
Capecitabine	2259	2273	1863	1752	1607	1769	1797	1899
Prednisone	608	627	635	658	693	783	899	963
Azathioprine	553	606	633	672	710	753	793	815
Megestrol	619	614	538	448	397	391	467	447
Imatinib	245	259	262	274	280	270	282	297
Abiraterone	-	-	52	107	177	226	228	266
Ciclosporin	330	321	306	295	292	280	271	245
Bicalutamide	303	302	263	245	223	202	185	166
Tamoxifen	133	136	135	143	150	149	144	142
Gemcitabine	97	89	87	87	92	92	92	94
Nilotinib	42	53	65	77	82	81	93	93
Sorafenib	73	73	63	64	65	66	82	80
Leflunomide	44	46	46	47	50	52	55	57
Cyproterone	72	70	60	56	53	50	50	49
Exemestane	58	57	54	51	51	50	48	48
Flutamide	259	209	153	119	99	78	67	47

**Table 2 molecules-27-03203-t002:** Cytostatic drugs with the highest values of PEC_wwtp_ and PEC_river_ (in ng L^−1^) in the period of 2013–2017 and mean values from PEC_wwtp_ and PEC_river_. Mycophenolic acid (MPA); Hydroxycarbamide (HYD); Capecitabine (CAP); Bicalutamide (BIC); Imatinib (IMA); Prednisone (PRED); Leflunomide (LEF); Abiraterone (ABI); Pazopanib (PZ); Paclitaxel (PTX); Azathioprine (AZA); Rituximab (RIX); Nilotinib (NILO); Megestrol (MEG); Cyproterone (CPA); Trastuzumab (TMAB); Pemetrexed (PMT); Flutamide (FLUT); Mercaptopurine (6-MP); Ifosfamide (IFO).

	2013			2014			2015			2016			2017		PEC_wwtp_ (mean ± sd)	PEC_river_ (mean ± sd)
	PEC_wwtp_	PEC_river_		PEC_wwtp_	PEC_river_		PEC_wwtp_	PEC_river_		PEC_wwtp_	PEC_river_		PEC_wwtp_	PEC_ri__ver_		
MPA	2224	85.8	MPA	2422	93.4	MPA	2546	98.2	MPA	2640	102	MPA	2750	106	2516 ± 203	97.0 ± 7.8
HYD	892	34.4	HYD	944	36.4	HYD	994	38.3	HYD	1051	40.6	HYD	1098	42.4	995 ± 82	38.4 ± 3.2
CAP	165	6.4	CAP	152	5.8	CAP	167	6.4	CAP	170	6.5	CAP	179	6.9	166 ± 10	6.4 ± 0.4
BIC	132	5.1	BIC	120	4.6	BIC	109	4.2	BIC	100	3.8	BIC	89.6	3.4	110 ± 17	4.2 ± 0.7
IMA	65.1	2.5	IMA	66.6	2.6	IMA	64.1	2.5	IMA	67.0	2.6	IMA	70.7	2.7	66.7 ± 2.5	2.6 ± 0.1
PRED	43.2	1.7	PRED	45.5	1.7	PRED	51.4	2.0	PRED	59.0	2.3	PRED	63.1	2.4	52.4 ± 8.5	2.0 ± 0.3
LEF	42.2	1.6	LEF	44.6	1.7	LEF	46.5	1.8	LEF	49.2	1.9	LEF	51.2	1.9	49.0 ± 3.6	1.2 ± 0.1
PZ	18	0.69	PZ	29.6	1.1	ABI	28.2	1.1	PZ	31.0	1.2	ABI	33.2	1.3	25.0 ± 7.7	0.97 ± 0.30
CPA	15.9	0.61	ABI	22.1	0.85	PZ	27.9	1.1	ABI	28.4	1.1	PZ	31.2	1.2	27.5 ± 5.5	1.0 ± 0.1
MEG	14.1	0.54	CPA	14.9	0.57	AZA	14.9	0.57	PTX	16.7	0.64	PTX	19.3	0.74	14.8 ± 3.4	0.57 ± 0.04
AZA	13.3	0.51	AZA	14.1	0.54	PTX	14.8	0.57	AZA	15.7	0.60	AZA	16.1	0.62	14.8 ± 1.2	0.57 ± 0.04
ABI	13.3	0.51	PTX	12.8	0.49	CPA	14.2	0.55	MEG	14.7	0.56	RIX	15.1	0.58	13.5 ± 2.3	0.52 ± 0.08
NILO	11.7	0.45	NILO	12.5	0.48	NILO	12.4	0.48	NILO	14.2	0.55	NILO	14.3	0.55	13.0 ± 1.2	0.50 ± 0.05
FLUT	10.8	0.41	MEG	12.5	0.48	MEG	12.3	0.47	CPA	14.2	0.55	MEG	14.1	0.54	13.5 ± 1.1	0.52 ± 0.04
PTX	10.4	0.4	PMT	10.4	0.4	PMT	11.8	0.45	RIX	11.9	0.46	CPA	13.9	0.54	14.6 ± 0.8	0.56 ± 0.06
			IFO	10.2	0.36	IFO	10.3	0.39	TMAB	11.8	0.45	TMAB	12.2	0.47	12.0 ± 0.3	0.46 ± 0.05
									PMT	11.6	0.45	PMT	10.6	0.41	11.1 ± 0.7	0.43 ± 0.01
												FLUT	nd	nd	10.8 ± 0.0	0.41 ± 0.02
												6-MP	10.4	0.4	10.4 ± 0.0	0.40 ± 0.00
												IFO	nd	nd	10.3 ± 0.1	0.38 ± 0.02

**Table 3 molecules-27-03203-t003:** Concentration of cytostatic drugs published in the literature (ng L^−1^). PECs vs. MECs ratios.

	Negreira 2013				Negreira 2014		Ferrando-Climent et al. (2014)				Gomez-Canela et al. (2014)				Julià Martin et al.		Franquet 2017	
Highest PECs Cytostatics	WWTP inf	PEC/ MEC	River Water (Surface)	PEC/ MEC	WWTP eff	PEC/ MEC	WWTP eff	PEC/ MEC	River Water (Surface)	PEC/ MEC	WWTP eff	PEC/MEC	River Water (Surface)	PEC/ MEC	River Water (Surface)	PEC/ MEC	River Water (Surface)	PEC/ MEC
Mycophenolic acid																	656	0.2
Capecitabine	27	6.2			7.7	21.6					<15	<11.1						
Imatinib	<180	<0.4	<180	<0.01														
Prednisone											<12	<4.4						
Azathioprine							<6.1	<2.4	<3.9	0.1			<3.9	<0.1				
Paclitaxel	4.4	3.4	<3.1	<0.2			<8.7	<1.7	<2.9	<1.7			<2.9	<0.2	<0.2	<2.9		
Cyproterone											<4.1	<3.6						
Megestrol											<3–20	<4.5–0.7					6	0.1
Ifosfamide	<2.0	<5	<2.0	<0.2	8.9	1.2	<1.3	7.9	<1.1	<0.3	<6	<1.7	<1.1	0.3	<1.7	0.2	13.9	0.03

**Table 4 molecules-27-03203-t004:** EC_50_ and LC_50_ for highest PECs and riks quocient (RQ).

ATC Group	Name	Organism	Test	References	Toxicity (mg L^−1^)	PEC_river_ ng L^−1^ (Mean) ± SD	RQ
L04AA06	Mycophenolic acid	*D. magna*	48 h, EC_50_	[32]	>100	97.08 ± 7.84	9.71 × 10^4^
L01XX05	Hydroxycarbamide	*D. magna*	Acute toxicity, 48 h, EC_50_	[33]	>100	38.42 ± 3.20	3.84 × 10^4^
L01BC06	Capecitabine	*D. magna*	LC_50_, 48 h	[34]	224	6.40 ± 0.39	2.86 × 10^5^
		*D. magna*	Reproduction 48 h, EC_50_	[1]	850		7.53 × 10^6^
L02BB03	Bicalutamide	*D. magna*	24 h (static), EC_50_	(AstraZeneca, 2006)	1	4.22 ± 0.66	4.22 × 10^3^
		*B. sunfish*	96 h (static), LC_50_		4		1.06 × 10^3^
L01XE01	Imatinib	*D. magna*	48 h, LC_50_	[34]	11.97	2.58 ± 0.08	2.16 × 10^4^
H02AB07	Prednisone	*D.magna*	48 h, EC_50_	[35]	108.1	2.02 ± 0.33	1.87 × 10^5^
L04AA13	Leflunomide	*D. magna*	48 h, EC_50_	[36]	>100	1.78 ± 0.13	1.78 × 10^5^
		*D.rerio*	48 h, LC_50_	[32]	17		1.05 × 10^4^
L01EX03	Pazopanib	nd				1.06 ± 0.21	
L02BX03	Abiraterone	nd				0.97 ± 0.30	
L04AX01	Azathioprine	*D.magna*	48 h, EC_50_	[37]	>100	0.57 ± 0.04	5.68 × 10^6^
L01CD01	Paclitaxel	*D.magna*	48 h, EC_50_	[38]	0.74	0.57 ± 0.04	7.68 × 10^4^
G03HA01	Cyproterone	*D.magna*	48 h, EC_50_	[37]	2.4	0.56 ± 0.06	2.35 × 10^4^
L02AB01	Megestrol	*D.magna*	48 h, LC_50_	[39]	5	0.52 ± 0.04	1.04 × 10^4^
L01FA01	Rituximab	nd				0.52 ± 0.08	
L01EA03	Nilotinib	nd				0.50 ± 0.05	
L01XC03	Trastuzumab	*D.magna*	48 h, EC50	[36]	369	0.46 ± 0.05	1.25 × 10^6^
		*B. sunfish*	96 h, LC_50_	[36]	10		4.60 × 10^5^
L01BA04	Pemetrexed	*D.magna*	48 h, EC_50_	[37]	462	0.43 ± 0.01	9.31 × 10^7^
		Fish (unknown)	96 h, LC_50_	[37]	1099.6		3.91 × 10^7^
L01BB02	Mercaptopurine	*D.magna*	48 h, EC_50_	[40]	55	0.40 ± 0.00	7.27 × 10^6^
L01AA06	Ifosfamide	*D.magna*	49 h, EC_50_	[41]	1795	0.38 ± 0.02	2.09 × 10^7^

## Data Availability

The data presented in this study are available on request from the corresponding author.

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
