# Peer review of "Long-Term Calculation of Predicted Environmental Concentrations to Assess the Risk of Anticancer Drugs in Environmental Waters"

_molecules, 2022, doi:10.3390/molecules27103203_

Round 1

Reviewer 1 Report

GENERAL COMMENTS

The manuscript “Long-term calculation of predicted environmental concentrations to assess the risk of anticancer drugs in environmental waters” from Dominguez-Garcia et al. deals with risk assessment of anticancer drugs in the region of Barcelona (Catalonia) in the years 2011 to 2017.

From a toxicological point of view all essential and necessary data are collected, but then the problem with the environmental data arises. From an environmental chemist’s view the following points have to be addressed.

The authors need information on all WTPs (primary, secondary, tertiary and potentially Advanced Oxidation Process) of Catalonia (effluent concentrations and budget calculations for a time span – week or even month).

The authors need information on the receiving river before the WTP effluent is introduced (concentrations of anticancer drugs and data from the river gauge for budget calculations).

The authors need information along all rivers taking samples from the first WTP effluent to the river mouth to the sea to get a valuable data set on the environmental behaviour of anticancer drugs.

All the available environmental data from anticancer drugs seems to me as cherry picking from

all over the world (mainly Europe) and is largely prone to a high span of error. Little is known about the quality of sampling.

Overall merit of the manuscript:

The toxicological approach to anticancer drugs in the environment is appreciated, but the predicted environmental concentrations (PEC) have to be questioned due a lack of reliable real data taken in environmental compartments. This manuscript might be an incentive for a larger sampling study taking into account the above mentioned critical points.

Despite my criticism I recommend a publication of the manuscript after major revision, if the issue of the used data basis is broached and the need for a comprehensive environmental study is stated and acknowledged.

Author Response

Reviewer 1

The manuscript “Long-term calculation of predicted environmental concentrations to assess the risk of anticancer drugs in environmental waters” from Dominguez-Garcia et al. deals with risk assessment of anticancer drugs in the region of Barcelona (Catalonia) in the years 2011 to 2017.

From a toxicological point of view all essential and necessary data are collected, but then the problem with the environmental data arises. From an environmental chemist’s view the following points have to be addressed.

The authors need information on all WTPs (primary, secondary, tertiary and potentially Advanced Oxidation Process) of Catalonia (effluent concentrations and budget calculations for a time span – week or even month).

The authors need information on the receiving river before the WTP effluent is introduced (concentrations of anticancer drugs and data from the river gauge for budget calculations).

The authors need information along all rivers taking samples from the first WTP effluent to the river mouth to the sea to get a valuable data set on the environmental behaviour of anticancer drugs.

All the available environmental data from anticancer drugs seems to me as cherry picking from all over the world (mainly Europe) and is largely prone to a high span of error. Little is known about the quality of sampling.

Thanks for your suggestion. We really think that all the information that the reviewer suggest is not necessary in this manuscript. The main aim of this study is to demonstrate the usefulness of PEC calculations for the long-term estimation of 132 anticancer drugs in wastewater and rivers over the period 2010-2017 and prioritize the most prevalent to be considered in monitoring studies. Moreover, for compounds with the highest PEC values, a risk assessment study has been conducted in order to predict their possible effects in the aquatic environment. It is very important to remark that all the values reported in the manuscript are estimated but it helps to visualize the possible environmental risks of these new class of contaminants.   

Overall merit of the manuscript:

The toxicological approach to anticancer drugs in the environment is appreciated, but the predicted environmental concentrations (PEC) have to be questioned due a lack of reliable real data taken in environmental compartments. This manuscript might be an incentive for a larger sampling study taking into account the above mentioned critical points.

Despite my criticism I recommend a publication of the manuscript after major revision, if the issue of the used data basis is broached and the need for a comprehensive environmental study is stated and acknowledged.

Thanks for your comments. We completely agree with the reviewer that there are many lacks of real data for anticancer drugs. However, this manuscript pretends to demonstrate the necessity of a complete monitoring of WWTPs and rivers of Catalonia for cytostatic compounds.

Reviewer 2 Report

My comments to the manuscript (ID_ molecules-1699046: Long-term calculation of Predicted Environmental Concentra- 2 tions to assess the risk of anticancer drugs in environmental 3 waters) were given below.

  1. It should be checked and should be followed strictly the guidelines of the journal. (Introduction, Results, Discussion, Materials and Methods, Conclusions). The section (Discussion) is not found in the manuscript.

  1. A graphical abstract should be provided.

  1. Data of (Water consumption (W, L inhabitant-1 day-1 ); ?ℎ?) should be provided.

  1. Detail explanation of using (DF used was following the value proposed by Keller et al.) should be provided. Is this DF value suitable to this work?

  1. Data for this work were from 2013 to 2017 period. Is it still value for this period (2022)? An explanation of old data and the meaning of this work should be provided in details.

Author Response

Reviewer 2

My comments to the manuscript (ID_ molecules-1699046: Long-term calculation of Predicted Environmental Concentrations to assess the risk of anticancer drugs in environmental 3 waters) were given below.

1. It should be checked and should be followed strictly the guidelines of the journal. (Introduction, Results, Discussion, Materials and Methods, Conclusions). The section (Discussion) is not found in the manuscript.

We have changed the subsection to “Results and discussion” in line 75 of the manuscript. This section englobes all the results and the discussion of them.

2. A graphical abstract should be provided.

A graphical abstract has been added to a different document as indicated summarizing the main information of the manuscript.

3. Data of (Water consumption (W, L inhabitant-1 day-1 ); ?ℎ?) should be provided.

The value of water consumption has been added in the manuscript (section 3.2). “…(about 130.9 L inhab-1 d-1 in Catalonia)…” in line 311.

4. Detail explanation of using (DF used was following the value proposed by Keller et al.) should be provided. Is this DF value suitable to this work?

In line 326 has been added a detail explanation of the use of DF value from Keller et al. study. Moreover, this value was used in the previous published paper (Franquet-Griell et al. 2015) and the present study is a continuation with the newest consumptions in 2013, 2014, 2015, 2016 and 2017.

5. Data for this work were from 2013 to 2017 period. Is it still value for this period (2022)? An explanation of old data and the meaning of this work should be provided in details.

In this study data was reported from 2013 to 2017 period. As explained before, this is a continuation work from the previous one published in 2015 (Franquet-Griell et al. 2015).  Contemporaneous data is intended to be treated when available to implement and widen PEC calculations of cytostatic compounds and obtain more information regarding environmental threats.

Round 2

Reviewer 1 Report

I think the authors should clearily emphasize within the abstract that their predictions on PEC of anticancer drug concentrations are not based on own measurements in wastewaters and rivers.

Reviewer 2 Report

The manuscript “Long-term calculation of Predicted Environmental Concentrations to assess the risk of anticancer drugs in environmental waters” was revised.

The authors have tried their best to address the comments of reviewers clearly. The revised manuscript is improved. It could be accepted to publish in the Molecules.